# Incidence of electrocardiographic and electrolyte changes in acute oleander poisoning in humans: A systematic review and meta-analysis protocol

Asanka Eriyawa[1,¤,*], Shaluka Jayamanne[2‡], Niroshan Lokunarangoda[3‡], Arunasalam Pathmeswaran[4‡], Rajeevan Francis[5‡], Kanagasingam Arulnithy[6‡], Dinuka Dharmapala[1], Pradeepa Jayawardane[7‡*]

1 Department of Pharmacology, Faculty of Medicine, Wayamba University of Sri Lanka, Kuliyapitiya, Sri Lanka, 2 Department of Medicine, Faculty of Medicine, University of Kelaniya, Ragama, Sri Lanka, 3 Department of Medicine, Faculty of Medicine and Mental Health, University of Moratuwa, Moratuwa, Sri Lanka, 4 Faculty of Medicine, University of Kelaniya, Ragama, Sri Lanka, 5 Department of Clinical Sciences, Faculty of Health Care Sciences, Eastern University of Sri Lanka, Batticaloa, Sri Lanka, 6 Teaching Hospital Batticaloa, Batticaloa, Sri Lanka, 7 Department of Pharmacology, Faculty of Medical Sciences, University of Sri Jayewardenepura, Nugegoda, Sri Lanka

☙ These authors contributed equally to this work.
‡ SJ, NL, AP, RF, KA, PJ also contributed equally to this work.
¤ Current Address: Faculty of Graduate Studies, University of Sri Jayewardenepura, Sri Lanka
* pradeepa@sjp.ac.lk (PJ); asankaeriyawa@wyb.ac.lk (AE)

## Abstract

Yellow oleander (*Cascabela thevetia*, previously known as *Thevetia peruviana*) & Common oleander (*Nerium oleander*) contain a wide range of cardiac glycosides and their toxicity is similar to digoxin toxicity. It causes arrhythmogenesis by directly and indirectly influencing cardiac myocytes. Oleander poisoning leads to high morbidity and mortality, causing considerable healthcare burden worldwide. This systematic review and meta-analysis aim to synthesize evidence from studies reporting the effects of electrocardiographic and biochemical changes following acute oleander toxicity. It will contribute to identifying the true incidence of electrocardiographic and electrolyte changes which will be useful in-patient management and clinical decisions. Electronic databases, Google Scholar, and reference lists of relevant articles will be searched. Human studies reporting electrolyte and electrocardiographic changes following acute yellow oleander and common oleander toxicity globally from 1988 onwards will be included. The eligibility of studies will be checked by two reviewers independently, and the risk of bias will be evaluated for each study using the JBI critical appraisal tool. The Rayyan software will be used for the systematic review while heterogeneity of studies will be assessed using $I^2$ statistics. In the case of $I2 > 50\%$, meta-analysis will be conducted using a random effects model using STATA software. Publication bias will be assessed using visual inspection of funnel plots & Egger's weighted regression. The incidence of electrocardiographic and biochemical changes following oleander poisoning will directly influence patient management and guide improvements in healthcare facilities. Although ethical approval is not required for the systematic review, it will be disseminated through peer-reviewed publications, index journals, and scientific conferences.

**Data availability statement:** No datasets were generated or analysed during the current study. All relevant data from this study will be made available upon study completion.

**Funding:** The author(s) received no specific funding for this work.

**Competing interests:** The authors have declared that no competing interests exist.

## PROSPERO registration

PROSPERO registration number for this protocol is CRD42023451171

## Introduction

Yellow oleander (*Cascabela thevetia*, previously known as *Thevetia peruviana*) & Common oleander (*Nerium oleander*) are categorized under the family of Apocynaceae which are abundant in tropical and subtropical parts of the world as they are cultivated as ornamental trees [1]. High amounts of cardiac glycosides including thevetin A & B, thevetoxin, peruvoside, ruvoside, and nerifolin are abundant in the kernel of seeds, leaves, and fruits of yellow oleander. Common oleander has oleandrin, digitoxigenin, folinerin and digitoxigenin in its seeds, leaves and fruits [2].

Human exposure to oleander could be due to deliberate self-harm attempts, accidental ingestion, or criminal ingestion. Oleander poisoning incidents were reported from South Asia, Europe, and the United States including Hawaii, Australia, South Africa, East Asia, and the Solomon Islands [2,3], making it a global neglected tropical disease. Oleander poisoning is considered to be the most common plant poisoning in some South Asian countries, which is mainly due to deliberate self-harm [4]. Large numbers of oleander seed ingestion cases are recorded each year in South Asia making it a considerable burden to healthcare services due to high morbidity and mortality[1,3]

The occurrence of arrhythmia is due to the action of cardiac glycosides on both the myocardium and conduction system of the heart and also due to the autonomic activity which is neutrally mediated. Cardiac glycosides inhibit $Na^+/K^+$ ATPase pump in the cardiac cell membrane causing hyperkalemia and an increase in intracellular calcium levels leading to cardiac muscle depolarization. The transient flow of ions into the cell makes the cell irritable and arrhythmogenic. Effect on the central sympathetic system also bears a major role in arrhythmogenesis. Direct effects on myocytes are caused by histamine, nitric oxide, and angiotensin-like mediators [5,6].

The clinical presentation of oleander poisoning is similar to that of digitalis toxicity. The commonest clinical features of oleander poisoning patients who developed substantial cardiotoxicity are nausea, vomiting, weakness, fatigue, diarrhea, dizziness, abdominal pain, and visual symptoms [7].

Cardiac glycosides cause vagotonic effects like bradycardia and heart block. It increases the atrioventricular (AV) nodal delay and reduces the atrial and ventricular refractory period. Bradyarrhythmia from acute oleander poisoning manifests as sinus bradycardia, AV nodal block, sinus arrest, junctional rhythms, first-degree, second-degree, and third-degree atrioventricular blocks. [8,9].

Biochemical changes include hyperkalaemia mainly hypokalaemia. Serum magnesium levels appeared to be normal in acute oleander poisoning. However, if hypomagnesemia is present, it may worsen cardiac toxicity and predispose to cardiac arrhythmia. Therefore, treating hypomagnesemia is important in acute oleander poisoning management. Hypokalemia is treated with intravenous potassium, as it may worsen the cardiac toxicity. Serum potassium levels are measured every 6 hours and Insulin-dextrose solutions are used to treat hyperkalemia [3,10].

The main cardiac effect of yellow oleander poisoning is bradyarrhythmia. Intravenous atropine and isoprenaline are used in medical management(8). The need for temporary cardiac pacing in second- and third-degree atrioventricular blocks has influenced doctors to transfer potential candidates to cardiac specialized centers, which takes a few hours without optimal monitoring of the patient [10].

Tachyarrhythmias are very rarely observed in acute oleander poisoning. They are considered to be more dangerous and difficult to treat compared to Brady arrhythmias. There were no studies have been performed to assess the effectiveness of anti-arithmetic drugs. Ventricular tachyarrhythmias are usually treated with lidocaine [11].

Gastric decontamination by single dose and multiple doses of activated charcoal. The effect of a single dose has been evaluated in two randomized controlled trials, with contradictory results. However multiple doses of activated charcoal are preferred by most clinicians as the decontamination method [10].

Digoxin-specific antibody fragments have been used as the definitive treatment to neutralize the toxin as the antidote. It is considered the most effective and safe treatment method for severe acute oleander poisoning cases [12]. However, due to financial constraints currently not used in resource-poor settings in some countries.

It is difficult to accurately determine the correct incident of electrocardiographic and biochemical changes in acute oleander poisoning when reviewing the existing literature. True incidents are critical to know as it is important for clinicians to make decisions to identify emergencies that require immediate attention, prioritize treatment strategies, and anticipate possible complications due to acute oleander poisoning.

Most of the available current studies are characterized by various demographics, geographical locations, toxin doses, and study designs with varying sample sizes. To get a broader and more representative understanding of the evidence it is essential to compile and synthesize all evidence through a systematic review and meta-analysis [13].

Management of acute oleander poisoning heavily relies on biochemical and electrocardiographic derangements [3,10]. Therefore, this systematic review and meta-analysis will give the true incident which will be vital for clinical decision-making and support evidence-based care.

## Objective

The objective of this meta-analysis is to systematically identify, evaluate, and synthesize evidence on the incidence of electrocardiographic and biochemical changes in acute oleander poisoning in humans, aiming to provide accurate and clinically relevant estimates within a defined timeframe, thereby facilitating clinicians with reliable data to identify significant changes and make evidence-based decisions for effective patient management across various healthcare settings.

In resource-limited settings, identifying the incidence of electrocardiographic and biochemical changes in acute oleander poisoning will aid in resource allocation by prioritizing high-risk patients for interventions such as temporary cardiac pacing, antidote administration, Close monitoring or transfer to tertiary care centers with advanced facilities, enabling timely clinical decision-making.

## Methods

The protocol was prepared according to the Joanna Briggs Institute (JBI) manual for evidence synthesis [14] and Preferred Reporting Items for Systematic Reviews and Meta-analysis Protocol (PRISMA-P) [15], which provides a guide to systematic reviews and meta-analysis. The protocol was registered on PROSPERO (CRD42023451171).

The planned use JBI critical appraisal tool will enhance transparency in the appraisal process and allow authors to decide on inclusion and exclusion studies based on methodological rigor [16]. The structured approach of the tool will help to synthesize high-quality evidence, improving the credibility and conclusion while reducing the bias. The tool also guides to interpretation of the results while assessing study reliability and validity [17].

This systematic review and meta-analysis protocol will contain eligibility criteria which include inclusion and exclusion criteria, Search strategy, and Study selection followed by an assessment of methodological quality, data extraction, statistical analysis, and evidence synthesis and discussion.

## Eligibility criteria

**Inclusion criteria.** Inclusion criteria were defined according to the CoCoPop mnemonic for systematic reviews of prevalence that stand for Condition, Context & Population to ensure clear and consistent inclusion criteria [18].

Condition: Studies that reported electrolyte changes and electrocardiographic changes in acute yellow oleander and common oleander toxicity will be selected.

Context: Studies that were conducted worldwide and patients managed in an inpatient setting will be selected.

Population: Studies conducted on all age groups will be selected.

Types of studies: In determining the incidence and prevalence, cross-sectional and cohort studies are the best suited among observational studies. Therefore, cross-sectional and cohort studies articles will be selected. Experimental studies that give incidence and prevalence data will be included. Articles published from 1988 to 2024 November will be included.

**Exclusion criteria.** Studies conducted on non-human subjects and studies published in languages other than English will be excluded.

Ensuring consistency in terminology and data interpretation is crucial in systematic reviews. Translating languages other than English (LOE) into English may introduce a bias, potentially affecting the accuracy of the results. Additionally, limited access to translation services is a challenge that must be addressed. However, it was identified that there would be a substantial bias in the studies when studies done in languages other than English were excluded from systematic reviews which will be a limitation of this meta-analysis [19].

**Search strategy.** The following databases will be searched: PubMed, ScienceDirect, MEDLINE, Web of Science, Embase, Scopus, EBSCO, and Cochrane Library. In addition, Google Scholar will be searched for potential studies. The backward literature search will be used to identify studies that may not be included in databases by going through reference lists of identified studies [20].

The following search terms will be identified in each database: 'Oleander', 'ECG changes', 'bradycardia', 'heart block', 'electrolytes', 'potassium', 'creatinine', and 'magnesium'. Special filters to search human studies and language filters will be applied. Search terms will be tailored according to the database. Assistance from the medical librarian will be obtained for the searching of databases.

Truncations, wildcards, proximity operators, Boolean operators 'AND' and 'OR' will be used where appropriate. The search strategy for PubMed is given below.

Search Query is (((((((((Oleander) AND (electrocardiographic changes)) AND (ECG changes)) AND (electrolyte)) AND (potassium)) AND (creatinine)) AND (magnesium))) AND (electrocardiographic changes)) OR (electrolyte changes)

Search detail is (("nerium"[MeSH Terms] OR "nerium"[All Fields] OR "oleander"[All Fields] OR "oleanders"[All Fields]) AND (("electrocardiographer"[All Fields] OR "electrocardiographers"[All Fields] OR "electrocardiographic"[All Fields] OR "electrocardiographical"[All Fields] OR "electrocardiographically"[All Fields] OR "electrocardiographics"[All Fields] OR "electrocardiography"[MeSH Terms] OR "electrocardiography"[All Fields] OR "electrocardiograph"[All Fields] OR "electrocardiographs"[All Fields]) AND ("change"[All Fields] OR "changed"[All Fields] OR "changes"[All Fields] OR "changing"[All Fields] OR "changings"[All Fields])) AND (("electrocardiography"[MeSH Terms]

OR "electrocardiography"[All Fields] OR "ecg"[All Fields]) AND ("change"[All Fields] OR "changed"[All Fields] OR "changes"[All Fields] OR "changing"[All Fields] OR "changings"[All Fields])) AND ("electrolyte s"[All Fields] OR "electrolytes"[MeSH Terms] OR "electrolytes"[All Fields] OR "electrolyte"[All Fields] OR "electrolytic"[All Fields] OR "electrolytical"[All Fields] OR "electrolytically"[All Fields]) AND ("potassium, dietary"[MeSH Terms] OR ("potassium"[All Fields] AND "dietary"[All Fields]) OR "dietary potassium"[All Fields] OR "potassium"[All Fields] OR "potassium"[MeSH Terms]) AND ("creatinin"[All Fields] OR "creatinine"[MeSH Terms] OR "creatinine"[All Fields] OR "creatinines"[All Fields]) AND ("magnesium"[MeSH Terms] OR "magnesium"[All Fields] OR "magnesium s"[All Fields] OR "magnesiums"[All Fields]) AND (("electrocardiographer"[All Fields] OR "electrocardiographers"[All Fields] OR "electrocardiographic"[All Fields] OR "electrocardiographical"[All Fields] OR "electrocardiographically"[All Fields] OR "electrocardiographics"[All Fields] OR "electrocardiography"[MeSH Terms] OR "electrocardiography"[All Fields] OR "electrocardiograph"[All Fields] OR "electrocardiographs"[All Fields]) AND ("change"[All Fields] OR "changed"[All Fields] OR "changes"[All Fields] OR "changing"[All Fields] OR "changings"[All Fields]))) OR (("electrolyte s"[All Fields] OR "electrolytes"[MeSH Terms] OR "electrolytes"[All Fields] OR "electrolyte"[All Fields] OR "electrolytic"[All Fields] OR "electrolytical"[All Fields] OR "electrolytically"[All Fields]) AND ("change"[All Fields] OR "changed"[All Fields] OR "changes"[All Fields] OR "changing"[All Fields] OR "changings"[All Fields]))

Filters: Humans, English, Publication year from 1988

The search strategy was developed using previous pilot searching, in consultation with experts in the toxicology field and keywords given in the relevant articles.

## Data management

Once the search is done, citation results will be imported to the Rayyan systematic review web application [21] for effective selection. Duplicate citations will be removed using the deduplication function. Zotero reference management software [22] will be used for citation management and organizing selected references.

## Study Selection

The titles and abstracts will be independently screened by the two reviewers about inclusion criteria. The Rayyan web-based system will be used in the screening process and the decisions of two reviewers will be blinded to each other [21]. Then full texts will be read by the reviewers to decide whether articles meet the inclusion criteria. Any uncertainty will be resolved through discussion with a third reviewer. Any additional information will be requested from the authors if questions arise about eligibility. The selected studies will be evaluated for methodological quality and the required data will be extracted using a specially designed data extraction sheet. We will use PRISMA flow charts to present the selection process describing the reasons for the exclusion of full-text obtained studies [23].

**Assessment of methodological quality.** JBI critical appraisal checklist for studies reporting prevalence data will be used by two of the authors independently to assess the quality of the study. It consists of 9 questions with 'yes', 'no', 'unclear', and 'not applicable' responses. Target population, sampling, sample size, study subjects, and data analysis will be assessed by each author individually. Any differences between the results of the two reviewers will be resolved following discussion with a third reviewer [14,16].

**Data extraction.** Two authors will independently perform data extraction on selected studies using data extraction forms for prevalence studies. The following data will be extracted.

Citation details: authors, title, journal, year, issue, volume, pages.

Generic study details: study design, country, setting, timeframe for data collection, study inclusion/exclusion information, measurement method, and main results (n/N).

During data extraction customized data extraction forms will be amended as required [16,18].

**Outcomes.** As the primary outcomes, death, discharge alive, temporary pacemaker insertion and cardioversion will be considered. As the secondary outcomes, high serum potassium levels, high serum creatinine levels, high serum magnesium levels, sinus bradycardia, reverse tick sign of digoxin toxicity, 1st-degree heart block, 2nd-degree heart block mobitz type 1, 2nd-degree heart block mobitz type 2, 3rd-degree heart block will be considered [9].

**Statistical analysis and evidence synthesis.** PRISMA guidelines will be used to represent results. Data from the systematic review will be represented in comprehensive tables and flowcharts. Graphs will be used where appropriate. Summarized quantitative results for individual studies will be presented with point estimates and interval estimates [13,24].

The heterogeneity of studies will be assessed using $I^2$ statistics. If $I^2 > 50\%$, meta-analysis will be performed using the random effects model. If $I^2 < 50\%$ fixed effect model will be used. Prevalence figures and 95% confidence intervals will be presented using forest plots [25] using STATA Software [26].

In case of missing data, an attempt will be made to contact original authors to obtain such data.

If meta-analysis is not possible, a systematic narrative synthesis will be presented from the findings of the included studies. The characteristics and findings of studies will be explained using text and summary tables. Publication bias will be assessed using visual inspection of funnel plots & Egger's weighted regression. $p < 0.1$ will be considered a statistically significant bias [25].

**Data Accessibility.** This Protocol was formulated in November 2024 and it will be set into implantation by the review team, following the guidelines specified in this document. No datasets were generated or scrutinized during the ongoing stage. Once the systematic review concludes, all relevant data from this review will be disclosed to the public.

Although ethical approval is not required for this review, The findings from the systematic review and meta-analysis will be shared through peer-reviewed publications, scientific symposia, and various platforms including conferences in the field of toxicology, which will enable accessibility to anyone. All the data used in this meta-analysis, including study details, and analysis results, will be shared in the published article, and any additional data can be fully available without restriction and also provided upon request to the corresponding author.

## Discussion

Acute poisoning due to yellow oleander and common oleander is common in most parts of the world, significant in the South Asian region. A better understanding of the incidence of electrocardiographic and biochemical changes in oleander toxicity will provide a clear understanding of the gravity of the problem and help in managing the healthcare burden. It will predict the development of atrioventricular blocks, therefore deciding the necessity of temporary cardiac pacing and the use of the antidote. It may ultimately contribute to improving clinical decisions and patient outcomes. To our knowledge, systematic reviews on this topic have not been published up to date to address the above research gap.

This paper outlines a protocol for a systematic review and a meta-analysis according to the PRISMA standards [15]. Any deviations from this protocol will be discussed in the final manuscript with justification. Dissemination of the findings will be done through scientific conferences and index journals.

The presence of data with high heterogenicity and missing data could be a potential major limitation of this systematic review and meta-analysis. If a high $I^2$ value with substantial heterogenicity is present, it will potentially make meta-analysis results unreliable. Sensitivity analysis will assess the robustness of results by excluding the studies with a high risk of bias [25]. By performing sub-group analysis, sources of heterogeneity such as variations of study design, population characteristics, and geographic location can be explored [27]. The impact of the study characteristics can be further quantified by meta-regression [28]. Sensitivity analysis will also assess the robustness of the results by excluding the research studies with missing data leading to a high risk of bias [25]. To improve statistical power, multiple imputation methods will estimate missing values depending on observed data [29]. Publication bias due to missing data can be adjusted by detecting trim and fill analysis [30]. Some individual studies may not report both biochemical and electrocardiographic data in a single study. If they are encountered, considered as the limitations of the study.

It is expected that the synthesized evidence on the incidence of electrocardiographic and biochemical changes in acute oleander poisoning will contribute to clinical decision-making, support evidence-based care, and make the basis for future studies related to oleander poisoning.

## Supporting information

**S1 File. PRISMA-P 2015 checklist.**
(DOC)

## Author contributions

**Conceptualization:** Asanka Eriyawa, Pradeepa Jayawardane, Shaluka Jayamanne, Dinuka Dharmapala.

**Methodology:** Asanka Eriyawa, Pradeepa Jayawardane, Shaluka Jayamanne, Niroshan Lokunarangoda, Rajeevan Francis, Kanagasingam Arulnithy, Dinuka Dharmapala.

**Supervision:** Pradeepa Jayawardane, Shaluka Jayamanne, Niroshan Lokunarangoda, Arunasalam Pathmeswaran.

**Writing – original draft:** Asanka Eriyawa, Dinuka Dharmapala.

**Writing – review & editing:** Pradeepa Jayawardane, Shaluka Jayamanne, Arunasalam Pathmeswaran.

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
