## [Decision Letter · Decision Letter 0]

20 Dec 2024

PONE-D-24-51489Incidence of electrocardiographic and electrolyte changes in acute oleander poisoning in humans: A systematic review and meta-analysis protocolPLOS ONE

Dear Dr. Jayawardane,

Thank you for submitting your manuscript to PLOS ONE. After careful consideration, we feel that it has merit but does not fully meet PLOS ONE’s publication criteria as it currently stands. Therefore, we invite you to submit a revised version of the manuscript that addresses the points raised during the review process.

We look forward to receiving your revised manuscript.

Kind regards,

Mariselvam Ramaiah, Dr

Academic Editor

PLOS ONE

Journal Requirements:

Additional Editor Comments:

Reviewer 1

The manuscript presents a systematic review and meta-analysis protocol focusing on the incidence of electrocardiographic (ECG) and electrolyte changes in patients with acute oleander poisoning. This is a timely and relevant topic given the significant health burden posed by oleander toxicity in many parts of the world. The methodology appears robust, adhering to established guidelines such as PRISMA-P and the JBI manual for evidence synthesis. Below are specific comments and suggestions for improvement.

Comments:

• The stated objectives should be specifying how the findings will directly impact clinical practice, particularly in resource-limited settings.

• The introduction section needs a brief discussion of existing gaps in the literature and how this review aims to address them.

• The introduction has more contexts on global incidence and relevance outside South Asia would enhance the generalizability of the review.

• The planned use of the JBI Critical Appraisal Tool is useful to specify how the findings from this tool will influence the synthesis and interpretation of the results.

• The discussion acknowledges limitations regarding heterogeneous data and missing information. However, strategies for mitigating these challenges (e.g., sensitivity analyses) should be expanded.

• Ethical approval is not required for this review, but the manuscript should emphasize plans for data sharing and transparency post-publication.

• To provide a rationale for excluding non-English studies and how this might impact results.

• Ensure consistency in referencing throughout the manuscript.

Reviewer 2

The authors have carried out regarding oleander poisoning. The review needs more references regarding the study of oleader poisoning. The data provided is also insufficient.

Reviewers' comments:

Reviewer's Responses to Questions

**Comments to the Author**

1. Does the manuscript provide a valid rationale for the proposed study, with clearly identified and justified research questions?

Reviewer #1: Yes

Reviewer #2: Partly

2. Is the protocol technically sound and planned in a manner that will lead to a meaningful outcome and allow testing the stated hypotheses?

Reviewer #1: Yes

Reviewer #2: Partly

3. Is the methodology feasible and described in sufficient detail to allow the work to be replicable?

Reviewer #1: Yes

Reviewer #2: Yes

4. Have the authors described where all data underlying the findings will be made available when the study is complete?

Reviewer #1: Yes

Reviewer #2: No

5. Is the manuscript presented in an intelligible fashion and written in standard English?

Reviewer #1: Yes

Reviewer #2: Yes

6. Review Comments to the Author

You may also provide optional suggestions and comments to authors that they might find helpful in planning their study.

Reviewer #1: The manuscript is generally well prepared, with a clear structure and adherence to guidelines. Addressing the above comments will enhance the clarity and impact of the protocol.

Reviewer #2: The authors have carried out regarding oleander poisoning. The review needs more references regarding the study of oleader poisoning. The data provided is also insufficient.

7. PLOS authors have the option to publish the peer review history of their article (what does this mean? ). If published, this will include your full peer review and any attached files.

**Do you want your identity to be public for this peer review?** For information about this choice, including consent withdrawal, please see our Privacy Policy .

Reviewer #1: No

Reviewer #2: No

---

## [Author Response · Author response to Decision Letter 1]

2 Feb 2025

2nd February 2025

Editor-in-Chief

PLOS ONE

Subject: Rebuttal Letter for Manuscript ID PONE-D-24-51489

Dear Editor,

We sincerely thank you and the reviewers for the constructive comments and suggestions provided for our manuscript titled Incidence of electrocardiographic and electrolyte changes in acute oleander poisoning in humans: A systematic review and meta-analysis protocol”. We have carefully reviewed all the feedback and made the necessary revisions to improve the quality of the manuscript. Below, we provide our point-by-point responses to the reviewers’ comments.

Reviewer 01

Reviewer Comment Revision Line Number

The stated objectives should be specifying how the findings will directly impact clinical practice, particularly in resource-limited settings The objective was rephrased and explained how the objective will directly impact clinical practice in resource-limited settings 115 -124

The introduction section needs a brief discussion of existing gaps in the literature and how this review aims to address them. A brief discussion of existing gaps in the literature is added to the introduction section while explaining how this review aims to address them. 102 -113

The introduction has more contexts on global incidence and relevance outside South Asia would enhance the generalizability of the review. Global incidence was added and generalizability was enhanced 55-62

The planned use of the JBI Critical Appraisal Tool is useful to specify how the findings from this tool will influence the synthesis and interpretation of the results. The use of the JBI critical appraisal tool in terms of the evidence synthesis and interpretation of the results was further explained. 130-134

The discussion acknowledges limitations regarding heterogeneous data and missing information. However, strategies for mitigating these challenges (e.g., sensitivity analyses) should be expanded. Heterogeneous data and missing information as potential limitations were discussed along with the strategies of mitigation used discussion section. 292-302

Ethical approval is not required for this review, but the manuscript should emphasize plans for data sharing and transparency post-publication. Plans for data sharing and transparency post-publication were added under the data accessibility section. 275-280

To provide a rationale for excluding non-English studies and how this might impact results. The rationale for using only studies in English was explained and it was identified as a limitation of the systematic review. 156-161

Ensure consistency in referencing throughout the manuscript. Added more references and ensured consistency of referencing throughout the protocol. 320-399

Reviewer 02

Reviewer Comment Revision Line Number

Language check with the help of some software Language check was done using Grammarly software

Check spelling and grammatical mistakes throughout the manuscript. Language check was done using Grammarly software

Revise the whole manuscript as per the journal guidelines.

Necessary formatting and syles were done as per the POLS ONE style templates.

Latest references must be used.

The review needs more references regarding the study of oleader poisoning. The data provided is also insufficient. New references with more research studies and more data were added and ensured consistency of referencing throughout the protocol. 320-399

In the Introduction part, Na+/K+ is changed into Na+/K+. Nessasry correction was made 65

Page No.4. Eligibility Criteria part – Verify the first line (according to CoCoPop). Verified the eligibility criteria and a reference was added. 142-143

The methodology should be precise and clear.

The methodology was re-arranged in a precise and clear manner 125-280

Have the authors described where all data underlying the findings will be made available when the study is complete? A ststement was added under data accesasibility section to ensure all data underlying the findings described in their manuscript fully available without restriction 275-280

In summary, we have incorporated all the reviewers’ suggestions to the best of our ability, and we believe the revised manuscript has been significantly improved as a result. A detailed list of changes is highlighted in the revised manuscript for your convenience.

We deeply value the opportunity to address these comments and appreciate your efforts in reviewing our work. Please feel free to contact us if further clarifications or revisions are needed.

Thank you for considering our revised manuscript for publication in PLOS ONE

Sincerely,

Dr. Asanka Eriyawa

1st Author (Corresponding Author)

Prof. Pradeepa Jayawardane

(Corresponding Author)

---

## [Decision Letter · Decision Letter 1]

18 Feb 2025

Incidence of electrocardiographic and electrolyte changes in acute oleander poisoning in humans: A systematic review and meta-analysis protocol

PONE-D-24-51489R1

Dear Dr. Jayawardane,

We’re pleased to inform you that your manuscript has been judged scientifically suitable for publication and will be formally accepted for publication once it meets all outstanding technical requirements.

Kind regards,

Mariselvam Ramaiah, Dr

Academic Editor

PLOS ONE

Additional Editor Comments (optional):

Accept

Reviewers' comments:

Reviewer's Responses to Questions

**Comments to the Author**

1. Does the manuscript provide a valid rationale for the proposed study, with clearly identified and justified research questions?

Reviewer #1: Yes

Reviewer #2: Yes

2. Is the protocol technically sound and planned in a manner that will lead to a meaningful outcome and allow testing the stated hypotheses?

Reviewer #1: Yes

Reviewer #2: Yes

3. Is the methodology feasible and described in sufficient detail to allow the work to be replicable?

Reviewer #1: Yes

Reviewer #2: Yes

4. Have the authors described where all data underlying the findings will be made available when the study is complete?

Reviewer #1: Yes

Reviewer #2: Yes

5. Is the manuscript presented in an intelligible fashion and written in standard English?

Reviewer #1: Yes

Reviewer #2: Yes

6. Review Comments to the Author

You may also provide optional suggestions and comments to authors that they might find helpful in planning their study.

Reviewer #1: Incidence of electrocardiographic and electrolyte changes in acute oleander poisoning in humans: A systematic review and meta-analysis protocol. The authors addressed the comments in the revised manuscript's clarity, methodology, referencing, and presentation improved for enhancing the manuscript’s quality.

Reviewer #2: The author have done extensive work and the work was well aligned. The data was presented in a clear manner. The changes have also been incorporated.

7. PLOS authors have the option to publish the peer review history of their article (what does this mean? ). If published, this will include your full peer review and any attached files.

**Do you want your identity to be public for this peer review?** For information about this choice, including consent withdrawal, please see our Privacy Policy .

Reviewer #1: No

Reviewer #2: No

---

## [Editor Report · Acceptance letter]

PONE-D-24-51489R1

PLOS ONE

Dear Dr. Jayawardane,

I'm pleased to inform you that your manuscript has been deemed suitable for publication in PLOS ONE. Congratulations! Your manuscript is now being handed over to our production team.

Kind regards,

on behalf of

Dr. Mariselvam Ramaiah

Academic Editor

PLOS ONE